# A Coarse-to-Fine Auto-Sampler For Long-tailed Image Recognition

**Tong Wu**
Department of Information Engineering
The Chinese University of Hong Kong
Shatin, Hong Kong
tongwu@link.cuhk.edu.hk

**Hao Li**
Department of Electronic Engineering
The Chinese University of Hong Kong
Shatin, Hong Kong
haoli@link.cuhk.edu.hk

## Abstract

The long-tail distributed data in the real world has always been a great challenge for deep learning. Current approaches to mitigate the long-tail issue include re-sampling and data augmentation. However, the hand-crafted re-sampling and augmentation strategies are sub-optimal. In this paper, we propose a coarse-to-fine auto-sampler that re-samples and augments imbalanced data automatically using reinforcement learning. The sampler consists of two parts: a patch sampler for augmenting data during representation learning, and a class-wise weighted sampler for classifier learning. Experiments on standard long-tailed datasets including CIFAR-10/100-LT and ImageNet-LTT shows the effectiveness of our method. We also conduct visualization to better demonstrate the policy of our agents. Our code will be made public before the final revision. Our video is available at https://drive.google.com/file/d/1oamWNap3rC3Ulyd5jSs1ZrZIQiBB-iR1/view?usp=sharing.

## 1 Introduction

Along with the wide adoption of deep learning, recent years have seen significant progress in visual recognition, especially the remarkable breakthroughs in classification tasks. However, real-world visual phenomena usually have a long-tail distribution [22, 7] and current approaches would suffer from a significant drop in performance due to the extremely imbalanced label distribution. A number of studies have been carried out recently [15, 10, 30] to address this important and challenging issue.

To mitigate the data imbalance issue, re-weighting and re-sampling are widely used, and different weighting and sampling strategies have been designed and proved to be effective [18, 6]. Recently, a decoupled training manner to separate the representation learning and classifier learning process has shown impressive advance. The former aims to learn an effective and robust feature embedding via instance-balanced data sampler together with various data augmentation schemes and the latter focuses on decision boundary adjusting through classifier fine-tuning with class-aware data sampler. In this work, we would show how reinforcement learning benefits the recognition task in both representation and classifier learning stages. On one hand, previous works mainly apply random re-sampling with pre-defined per-class ratios, e.g., the reciprocal of the data amount ratios, and it is reasonable to assume that these manually designed samplers may not be the optimal solutions and the generalization among different datasets is not decent. One the other hand, apart from the traditional data augmentation methods such as random crop and flip, mixup [28] and its variants [23, 27] have shown advanced performance as a regularization strategy, while the image-pairs or patches are usually selected in a random manner, which is reasonable to be further improved and we again leverage reinforcement learning to design a better patch sampler network in this paper.

Overall, we propose a two-level auto-sampler, namely coarse-to-fine auto-sampler, during the whole training process where the policies are optimized with reinforcement learning methods. It consists of

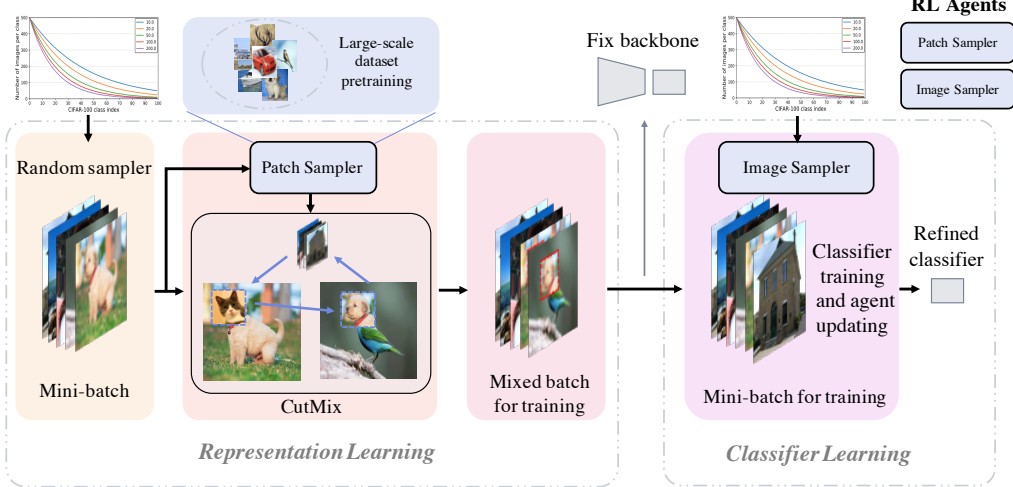

Figure 1: The framework of our method: a coarse-to-fine auto-sampler including an image-level per-class weighted sampling and a patch sampler to enhance the mixup process, with a decoupling training scheme.

two parts: an image-level per-class weighted sampling and a patch sampler to enhance the mixup process. They can be applied and evaluated separately or integrated into the decoupled learning scheme. Our framework is shown in Fig 1

We conduct experiments on several long-tailed recognition datasets to evaluate the effectiveness of our method. Furthermore, extensive comparison among searched and manually designed sampling weights as well as the performance between RL-based and random patch sampler would be included in the ablation studies.

## 2 Related Works

### 2.1 Reinforcement learning in deep learning tasks

There are already a number of successful applications of RL to the deep learning training process. For example, learning a sequence of operations for data augmentation from raw data with **AutoAugment** [4] and its improved and accelerated approaches [5, 13, 29]; Latest works have also explored the application of RL to adaptive **meta-weighting** [19] and **patch sampler** [3] in few-shot learning. These would be the most important references for us to formulate our problem and design the framework.

### 2.2 Long-tailed recognition

Current methods to address the issue of long-tailed recognition mainly include **re-sampling** [1, 18], **cost-sensitive learning** [14, 6], **transfer learning** [15], and the **decoupling of representation of classifiers** [30, 10]. Among them, re-sampling is a simple yet effective strategy that can be applied alone or along with other techniques, while the downsides of it are also well studied, including over-fitting, missing important samples, and hurt representation learning. Some latest works proposed to combine random sampling and class-balanced sampling with split network structure or training stage, while we aim to directly learn a specific sampling scheme that is adaptive to data distribution and training techniques (*e.g.* Mixup and metric learning) with RL as a powerful tool.

### 2.3 Sampling schemes in various approaches

The most widely used sampling rule is random sampling, which is simple, straight-forward, and effective for many deep learning tasks. While for some specific tasks or scenes, it may not be the most satisfying way. For example, **Deep metric learning** [8, 16, 20, 24] aims to learn an embedding space where inner-class distance is minimized while intro-class distance is maximized. The construction of

**Algorithm 1:** Optimizing per-class sampling weights

---

**Input :** Initialized network $f(\cdot; \theta_0)$, initialized distribution $\mu_0$ and $\sigma^2$, training set $\mathcal{D}_{\text{train}}$ and validation set $\mathcal{D}_{\text{val}}$

**Result:** Obtained optimal policy $\mu^*$

---

**for** $t = 1$ **to** $T$ **do**

    **for** $m = 1$ **to** $M$ **do**

        Sample class weights $W_m^{(t)} \sim G(\mu_t, \ \sigma^2 I)$;

        Network training $\theta^*(W_m^{(t)}) = \arg\max_\theta L(\mathcal{D}_{\text{train}}; \theta)$, with $\theta$ initialized from $\theta_0$;

        Compute the reward $r(W_m^{(t)}) = \text{Acc}(\theta^*; \mathcal{D}_{\text{val}})$;

    **end**

    Update $\mu_{t+1} = \arg\max_\mu \frac{1}{M} \sum_{m=1}^M R(\mu, \mu_t, W_m^{(t)})$;

**end**

**return** $W^* = \arg\max_{W_m^{(t)}} r(W_m^{(t)}), \forall t = 1, \ldots, T$

---

samples in a mini-batch is important to form effective metric learning pairs []. **Mixup** [28, 23, 27] is also a promising data augmentation method, while previous works mainly focus on large-scale and balanced datasets and the sampling of mixed instances is basically random. We would address the issue of data distribution and class relation in the RL-based policy design of the sampler. **Importance sampling** [11, 9] is also a related area for our reference.

## 3 Method

In this section, we first formulate the long-tailed image recognition problem in a unified notation set. Then we introduce the per-class weighted sampling and class-aware data augmentation separately in the reinforcement learning framework.

### 3.1 Problem Formulation

Suppose the dataset is divided into three subsets: training set $\mathcal{D}_{\text{train}}$, validation set $\mathcal{D}_{\text{val}}$ and test set $\mathcal{D}_{\text{test}}$, composed by images $I$ with labels $y$ from $C$ classes. Let $N_{\text{train}}$, $N_{\text{val}}$ and $N_{\text{test}}$ be the number of images in each subset. The mini-batch SGD is employed for optimization. During training, for the $k$-th step, a mini-batch $\mathcal{B}_k = \{I_{k_1}, I_{k_2}, ..., I_{k_B}\}$ is sampled from the training set, and is fed into a neural network $f(I; \theta)$ with parameters $\theta$ for generating predictions $\hat{y}$. After that, a loss function $L(I, y)$ is computed, whose gradients are propagated backward through the network to update the weights $\theta$.

### 3.2 Per-Class Weighted Sampling

In the standard SGD, the mini-batches are constructed by uniformly random sampling over the training set $\mathcal{D}_{\text{train}}$. However, due to the class imbalance in the long-tailed datasets, many few-shot classes are hardly sampled, and therefore hurting the performance of the classification. Weighted sampling is proposed to increase the probability of being sampled for the few-shot classes. Formally, the mini-batches are sampled from each class $c = \{1, 2, ..., C\}$ with a probability

$$p(c|\mathcal{D}_{\text{train}}) = w_c. \tag{1}$$

In the standard SGD, the probabilities are set as the class frequency

$$w_c = \frac{N_{\text{train}}^{(c)}}{N_{\text{train}}},$$

where $N_{\text{train}}^{(c)}$ is the number of images belonging to class $c$ in the training set.

In weighted sampling, the probabilities are defined to pay more attention to the few-shot classes. A common practice is to re-weight the different classes to make every class has the same probability of

being sampled.

$$w_c = \frac{1}{C}.$$

The manually designed weights are sub-optimal. We propose to optimize the probabilities $w_c$ with the PPO2 [17] algorithm. Specifically, following DDPG [12], in each trajectory, we sample $M$ groups of weights $W_m = \{w_c\}_m, m = 1, ..., M$ from a Gaussian distribution $G$

$$W_m \sim G(\mu, \sigma^2 I), \tag{2}$$

where $\mu = \{\mu_c\}_{c=1}^M$ and $\sigma^2 I$ are the mean value and the covariance of the Gaussian distribution, respectively. In order to make sure that $\sum_{c=1}^C w_c = 1$, we apply a softmax function on the sampled $w_c$. By considering $\mu$ as the parameter of the policy and the sampling from $G(\mu, \sigma^2 I)$ as the actions of the RL agent, optimizing $W$ can be formulated as an reinforcement learning problem with only one state.

The optimizing algorithm is as follows. In the $t$-th trajectory, the sampled weights $W_m, m = 1, ..., M$ are used for training $M$ different networks $f(I; \theta_m)$. After the training, the networks $f(I; \theta_m^*)$ are evaluated on the validation set $\mathcal{D}_{\text{val}}$ to get the top-1 accuracy, which is used as the values of the state-action pair

$$r(W_m) = \text{Acc}(\theta_m^*; \mathcal{D}_{\text{val}}). \tag{3}$$

Following the PPO2 algorithm, the policy is updated by

$$\mu_{t+1} = \arg\max_\mu \frac{1}{M} \sum_{m=1}^M R(\mu, \mu_t, W_m), \tag{4}$$

where the reward $R(\mu, \mu_t, W_m)$ is computed as

$$R(\mu, \mu_t, W_m) = \min\left( \frac{p(W_m; \mu, \sigma^2 I)}{p(W_m; \mu_t, \sigma^2 I)} r(W_m), \text{ CLIP}\left( \frac{p(W_m; \mu, \sigma^2 I)}{p(W_m; \mu_t, \sigma^2 I)}, 1 - \epsilon, 1 + \epsilon \right) r(W_m) \right),$$

where $\min(\cdot, \cdot)$ picks the smaller item from its inputs, $\text{CLIP}(x, 1 - \epsilon, 1 + \epsilon)$ clips $x$ to be within $1 - \epsilon$ and $1 + \epsilon$, and $p(W_m; \mu, \sigma^2 I)$ is the PDF of the Gaussian distribution. Note that the mean reward of the $M$ samples is subtracted when computing $r(W_m)$ for better convergence. After $T$ trajectories, the $W_m$ with the highest value on the validation set is picked as the final policy. Algorithm 1 describes our algorithm for optimizing per-class weights.

### 3.3 Region-Aware Mixup

#### 3.3.1 Mixup Framework

The re-sampling approach generates redundancy and is easy to overfit to the rare classes [26]. Inspired by [3] which designs a data augmentation method for the few-shot learning problem and the success of mixup [28, 23, 27] in ImageNet Challenge, we consider using a region-aware mixup strategy to improve the representation learning.

Different from most data augmentation methods such as random crop, decolorization and rotation, which apply transforms on a single image, mixup [28] proposes to combine two random images from the training set to get the augmented data via

$$\begin{aligned} \tilde{I} &= \lambda I_i + (1 - \lambda) I_j, \\ \tilde{y} &= \lambda y_i + (1 - \lambda) y_j, \end{aligned} \tag{5}$$

which serves not only as an augmentation trick but also regularization of the CNNs. Beyond the original *input mixup* in Eq. 5, other widely used strategies include *manifold mixup* [23] that performs feature-level interpolation, CutMix [27] that combine training samples with binary mask, and many other variants. In this work, we adopt the idea of CutMix that cut out patches from one image and fill it into another to perform data augmentation:

$$\begin{aligned} \tilde{I} &= M \odot I_i + (1 - M) \odot I_j, \\ \tilde{y} &= \lambda y_i + (1 - \lambda) y_j, \end{aligned} \tag{6}$$

where $M \in \{0, 1\}^{W \times H}$ denotes a binary mask indicating where to drop out and fill in from two images. Our difference with them is that the patch is not randomly selected, but guided by a patch sampler network $g(I; \phi)$. Specifically, we follow Wang *et al* [25] to use an agent driven by reinforcement learning that aims to find out the most discriminative local part of the image. We would briefly introduce how it works in the following part.

### 3.3.2 Patch Sampler

Motivated by the fact that not all regions in an image are task-relevant, Glance and Focus Network (GFNet) [25] was proposed as a two-stage framework that perform the region selection operation by a sequential decision process. *Glance* means use the down-sampled(*e.g.*, $96 \times 96$) full image as the initial step to produce a quick prediction due to the small image size; and if the glance step does not give a confidence score that is high enough, a *Focus* step would be needed which proceeds progressively with iteratively localizing and processing the class-discriminative image regions, facilitating early termination in an adaptive manner.

**Network architecture.** The GFNet consists of four components: a global encoder $f_G$, a local encoder $f_L$, a classifier $f_C$, and the patch proposal network $g$. Both encoders are deep CNNs that extract deep representations from the inputs. They share the same network structure but different parameters, dealing with the down-sampled whole image or selected patch, respectively. The classifier is a recurrent network that aggregates the information from all previous inputs and produces a prediction at each step. The patch proposal network, also called patch sampler in our case, is another recurrent network that determines the location of each image patch. Please refer to their paper [25] for more details.

**Rewards and optimization** During training, after obtaining the action $l_{t+1}$ at $t^{th}$ step, the image is croped accordingly to get the next patch input $x_{t+1}$ and produce the prediction $p_{t+1}$. Then the patch proposal network receives a reward $r_{t+1}$ for the action $l_{t+1}$, which is defined as the increments of the softmax prediction probability on the ground truth labels, *i.e.*, $r_{t+1} = p_{t+1,y} - p_{t,y}$. And then, the goal of $g$ is to maximize the sum of the discounted rewards:

$$\max_{\theta} \left[ \sum_{t=2}^{T} \gamma^{t-2} r_t \right]. \tag{7}$$

Extensive training strategies are reported in the original paper, and we refer to the officially released code for implementation.

### 3.3.3 Patch Sampler driven CutMix

Now we illustrate how to make use of the patch sampler and combine it with Mixup strategy. It should be clarified that the agent does NOT have to perform with the same category set, which means that we can leverage any large-scale dataset to train the agent and use it as a universal (the domain gap should not be too large) saliency patch producer. So that once the agent is well-trained, its parameters are not updated along with the new datasets and ONLY the forwarding/inference process is performed.

Concretely, suppose the network take $T$ steps and produce a confidence score for each category at each time, then we first take the mean average of each column of the $T \times \hat{C}$ confidence matrix. Notice that $\hat{C}$ denotes the class number of the dataset used to train the patch sampler agent. We further take the argmax of the output $\hat{C}$-length score vector, extract the corresponding column from the original matrix and then take the argmax of the $T$-length vector. The first step above aims to find the most likely-to-be-true class predicted by the network, even though the exact label from $Set_C$ may not exist in $Set_{\hat{C}}$, the prediction approximately illustrates a discriminative area.

Given the coordinates for each image in a mini-batch from the patch sampler, the image where a patch is cut off from would directly adopt the coordinates, while the image that a patch would be filled into would take the most *unimportant* part and fill it with the newcomer. We simply take the vertex of the $224 \times 224$ image that has the largest distance to the center of the 96 patch, which we consider would , to the largest extent, reserve the information of the original image.

Finally, the soft-label strategy in Eq. 6 would correspondingly be formulated as below following [27]:

$$\lambda = (S_{patch}/S_{image})^{\tau}, \tag{8}$$

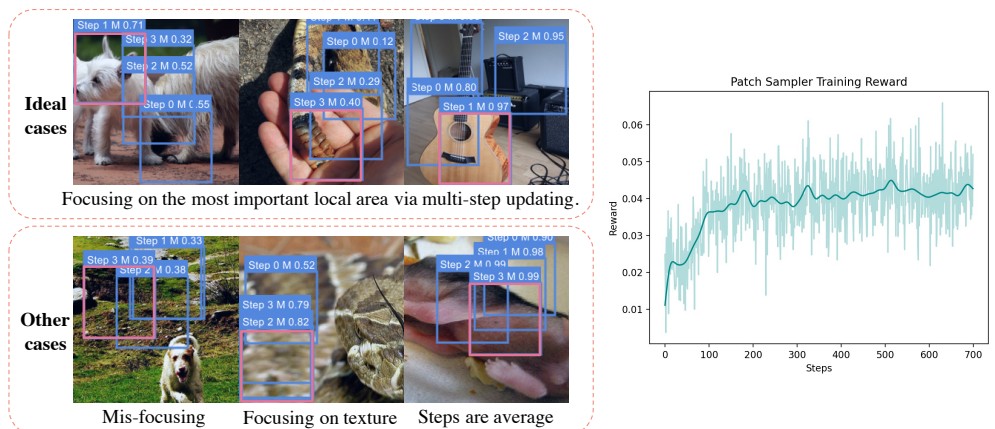

Figure 2: **Left:** Patch sampler performance. The first row shows examples of ideal performance, where the patch would focus on the discriminative part of the image; the second row shows some other cases: 1) the patch does not focus on the important area, 2) the whole image is filled with the object and the sampler could only select a local texture, and 3) the prediction score over different time-steps are average. **Right:** Reward curve during training.

where $\tau$ is a hyper-parameter. The original paper takes $\tau = 1$ while in our case we observe that taking $\tau \in [0.3, 0.5]$ produces the best performance. And since the image and patch size are fixed, we can simply defined $\lambda \in [0, 1]$ rather than considering $\tau$.

The patch sampler effectively selects the discriminative parts in one image and fill it to another while avoiding cutting off the important part in the latter.

### 3.4 Integration

Recently, it has been proposed that to decouple the learning of representation and classifier is beneficial for long-tailed recognition [10, 30]; thus a natural way to integrate the techniques above is to perform them in an overall two-stage manner: In the first stage, a uniform sampling strategy is applied with patch sampler guided CutMix, and we aim to enhance the representation learning by taking advantage of the regularization effect provided by the mixup strategy together with soft label; in the second stage, we freeze the network parameters of the feature extractor, and only fine-tune the classifier, a fully connected layer, using the searched sampling per-class weights. By doing this, we are actually alleviating the prediction bias via quick adjustment of the prediction boundary.

### 3.5 Proxy Tasks for Search

Searching for optimal per-class sampling weights and patch sampler network parameters are considerably time-consuming, due to the optimization of the classification network in each sampling. In order to reduce the search time, we use a light-weight proxy task during search. Specifically, we fine-tune the classifier of a ResNet-18 for one epoch, and evaluate on the validation set. Compared with the full training scheme with ResNet-50, the proxy task is 100x times faster.

## 4 Experiment

### 4.1 Datasets

The widely used datasets for long-tailed recognition includes CIFAR10-LT, CIFAR100-LT [6], ImageNet-LT [15] and iNaturalist [10]. We evaluate the per-class weighted sampling with CIFAR10-LT and CIFAR100-LT. And since the image size of CIFAR ($32 \times 32$) is too small for the patch sampler while our computation resources cannot afford large-scale experiments on ImageNet-LT, we further construct a little long-tailed dataset based on the ImageNet data, namely ImageNet-LLT. Concretely, it contains 10k images from 400 categories, the samples per-class ranging from 50 to 400.

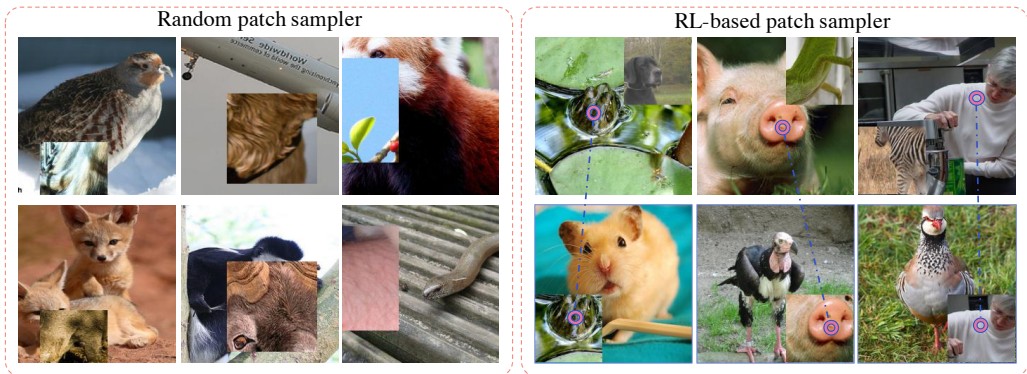

Figure 3: The comparison of random patch sampler and RL-based patch sampler. **Left, random patch sampler:** 1) choose the same size and location for all images in a mini-batch, where a column here comes from one mini-batch; 2) It does not select the important part of an image, thus most cut off parts are not class-discriminative; 3) It does not tend to avoid the occlusion issue when fill the patch into another image, thus may be placed the important locations of the latter that cause loss of information. **Right, RL-based patch sampler:** Global images with the discriminative local patch filled in to another image.

## 4.2 Experiments

Table 1: Experimental results on CIFAR-10-LT and CIFAR-100-LT

| Dataset | | CIFAR-10-LT | | CIFAR-100-LT | |
|---|---|---|---|---|---|
| **Imbalance Ratio** | | **100** | **50** | **100** | **50** |
| baseline | Vanilla model | 69.8 | 75.2 | 38.3 | 42.1 |
| sampling | random under-sampling | 65.9 | 73.1 | 32.8 | 39.0 |
| | random over-sampling | 66.8 | 73.5 | 33.0 | 38.9 |
| | class-balanced sampling | 69.6 | 76.0 | 32.7 | 38.5 |
| | square-root sampling | 68.6 | 75.2 | 35.5 | 40.2 |
| | progressively-balanced sampling [10] | 67.1 | 75.0 | 38.6 | 42.9 |
| SOTA | Focal Loss [14] | 70.4 | 76.7 | 38.4 | 44.3 |
| | Class-balanced loss [6] | 74.6 | 79.3 | 39.6 | 45.2 |
| | LDAM [2] | 77.0 | 81.0 | 42.0 | 46.6 |
| | BBN [30] | 80.6 | 83.6 | 42.6 | 47.0 |
| | TDE [21] | 80.6 | 83.6 | 44.1 | 50.3 |
| **Ours** | | 71.4 | 77.3 | 33.9 | 40.0 |

**Experimental results of weighted sampler.** Currently, we present the experimental results of a number of comparison methods including: different sampling strategies and the state-of-the-art long-tailed recognition methods. These results are shown in Table 1. Our sampler outperforms the baseline model and the common-used class-balanced sampling. On CIFAR-10-LT, our method outperforms Focal Loss. It is worth noted that on CIFAR-100-LT our method is not as well as on CIFAR-10-LR. The reason of the performance downgrade may be our search schedule is too short. On CIFAR-100-LT, there are 100 weights to be searched, which is harder than CIFAR-10-LT. We only trained for 100 trajectories due to resource limit, and longer training may bring better results on CIFAR-100-LT.

**Experimental results of patch sampler driven CutMix.** As shown in Table 2, where $_c$ denotes the probability of whether to apply mixup for the coming batch and $\lambda$ is the fixed soft-label proportion in Eq. 6. It can be seen that CutMix with random sampler gains improvement over the baseline; CutMix with RL-based patch sampler gains even better results, addressed in bold font. And the effect of different hyper-parameters are evaluated in the table.

Table 2: Top-1 and top-5 accuracy on ImageNet-LLT with different training strategies.

| Dataset | | | ImageNet-LLT | |
|---|---|---|---|---|
| **params** | $p_c$ | $\lambda$ | **Top-1** | **Top-5** |
| baseline | 0 | - | 50.59 | 75.13 |
| random mixup | 0.1 | - | 53.08 | 76.95 |
| | 0.3 | - | 51.01 | 75.53 |
| | 0.5 | - | 51.35 | 76.10 |
| | 0.7 | - | 50.47 | 75.87 |
| patch sampler mixup | 0.1 | 0.5 | 51.63 | 76.18 |
| | 0.3 | | 53.37 | 77.64 |
| | 0.5 | | 52.76 | 78.15 |
| | 0.7 | | **54.39** | **78.93** |
| | 0.5 | 0.1 | 51.25 | 77.32 |
| | | 0.3 | 52.29 | 77.21 |
| | | 0.5 | 52.76 | 78.15 |
| | | 0.7 | 53.21 | 77.33 |

And finally, the overall results with integral techniques outperforms the baseline by a large margin, as shown in Table 3.

Table 3: Overall experimental results of integrated techniques.

| Dataset | ImageNet-LLT | |
|---|---|---|
| **methods** | **Top-1** | **Top-5** |
| baseline | 50.59 | 75.13 |
| + Patch Sampler CutMix | 54.39 | 78.93 |
| + Image Sampler fine-tune | 60.61 | 80.32 |

### 4.3 Visualization

**Comparison between patch samplers.** We visualize more examples of patch sampler driven CutMix and comparison to random CutMix in Fig. 3 and show that the patch sampler effectively select the discriminative parts in one image and fill it to another while avoiding cutting off the important part in the latter.

**Several cases of CutMix combination.** We show some typical cases of CutMix. The source and target images could be the same in rare cases, could be from similar classes, or totally different classes, as shown in Fig. 4.

## 5 Conclusion

In this work, we propose a coarse-to-fine auto-sampler with a decoupling training scheme. The image-level per-class weighted sampling and a patch sampler to enhance the mixup process optimized with reinforcement learning methods can be applied and evaluated either separately or integrally. Experimental results show prove the effectiveness of our method.

# 6 Acknowledgement

We refer to several officially released code bases, namely OLTR [15] [1], GFNet [25] [2], and Cut-Mix [27] [3], to implement our experiments.

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
