# OpenReview forum: "A Coarse-to-Fine Auto-Sampler For Long-tailed Image Recognition"
_CUHK.edu.hk/2021/Course/IERG5350_

### Official Review · AnonReviewer2 · 2020-12-15
**RL-based sampler for Long-tailed Image  Recognition**

**Rating:** 7
**Confidence:** 4

**Review:**

Summary: This paper proposes a coarse-to-fine RL-based auto-sampler that includes 1) a patch sampler for augmenting data and 2) a class-wise weighted sampler for classifier learning. Both phases resort to the RL approach to eliminate hand-crafted/random policies. Experiments on CIFAR-10/100-LT and ImageNet-LTT shows the effectiveness of our method, compared to baseline methods.

Overall this paper is well-written and easy to follow with slight grammar mistakes. It is suggested to make the problem formulation more clear in the RL-based Patch Sampler part to make the paper more self-consistent (e.g., used algorithms, state/action representations).

The adoption of the RL approach is well-motivated, and the claims are mostly supported. The inclusion of RL approaches in both path sampler and weighted sampler is novel.

My main curious lies in the performance of RL-based weighted sampler, as compared to previous pre-defined per-class ratios (e.g., the reciprocal of the data amount ratios). While the author has listed the performance comparison in terms of accuracy, it may give more insights if the author could provide further analysis of the RL learned policies (e.g., on the 10 weights of CIFAR-10-LT), compared to fixed weights.

---

### Official Review · AnonReviewer1 · 2020-12-20
**Very good submission.**

**Rating:** 8
**Confidence:** 4

**Review:**

This paper focuses on the long-tailed data distribution problem. The contribution of this paper is to propose a coarse-to-fine auto-sampler that samples and augments imbalanced data with RL. The sampler consists of a path sampler and a class-wise weighted sampler. Experiments show that the proposed method has effectively improved the performance on CIFAR-10/100-LT and ImageNet-LTT. Visualization results also show better policies.

Advantages:
(1) The writing is clear. The overall paper is very easy to follow.
(2) The mathematical formulation and notations are clear and easy to understand.
(2)  Experiments show the effectiveness of the proposed method.

Questions:
(1) In Section 3.3.1, the authors introduce Mixup first, but then show that they use CutMix instead. It is unclear to me why the authors choose CutMix rather than the original Mixup. Are there any experiments that show the superiority of Cutmix?
If Mixup is also fine, is the region-aware design useless because Mixup involves the whole image?
(2) In Figure 1, the CutMix part shows the cat head is exactly placed upon the dog head. It is also similar to the bird image. The visualization performance here is too ideal.

---

### Official Review · AnonReviewer3 · 2020-12-20
**A novel auto-sampler for long-tailed image recognition**

**Rating:** 8
**Confidence:** 3

**Review:**

Summary: This paper focuses on processing imbalanced data automatically using reinforcement learning. The main contribution of this paper is the design of a coarse-to-fine auto-sampler.

Technical quality: The paper's technical quality is good. The pipeline is clearly defined. It would be better if detailed hyper-parameter setting of reinforcement learning method is mentioned.

Clarity: This paper is clearly written and the sections are organized in a logical way. Figures and tables in the paper are clear and informative.

Suggestion: I suggest the authors to add some new metrics to verify the performance of the proposed sampler.